# Cyclooxygenase-2 Activity Regulates Recruitment of VEGF-Secreting Ly6C^high^ Monocytes in Ventilator-Induced Lung Injury

**DOI:** 10.3390/ijms20071771

**Published:** 2019-04-10

**Authors:** Tzu-Hsiung Huang, Pin-Hui Fang, Jhy-Ming Li, Huan-Yuan Ling, Chieh-Mo Lin, Chung-Sheng Shi

**Affiliations:** 1Department of Respiratory Therapy, Chang Gung Memorial Hospital, 6, Sec West, Chia-Pu Road, Puzi, Chiayi County 61363, Taiwan; j55812633122@yahoo.com.tw (T.-H.H.); cj04m06@gmail.com (H.-Y.L.); 2Graduate Institute of Clinical Medical Sciences, College of Medicine, Chang Gung University, No. 259, Wenhua 1st Rd., Guishan Dist., Taoyuan City 33302, Taiwan; t18825131@yahoo.com.tw; 3Department of Emergency Medicine, National Cheng Kung University Hospital, College of Medicine, National Cheng Kung University, Tainan 704, Taiwan; fph2005er@gmail.com; 4Division of Colon and Rectal Surgery, Department of Surgery, Chang Gung Memorial Hospital, Puzi, Chiayi County 61363, Taiwan; 5Division of Pulmonary and Critical Care Medicine, Chang Gung Memorial Hospital, 6, Sec West, Chia-Pu Road, Puzi, Chiayi County 61363, Taiwan; 6Department of Nursing, Chang Gung University of Science and Technology, Chiayi Campus, Puzi, Chiayi County 61363, Taiwan; 7Division of Urology, Department of Surgery, Chang Gung Memorial Hospital, Puzi, Chiayi County 61363, Taiwan

**Keywords:** cyclooxygenase-2, Ly6C^high^ monocytes, ventilator-induced lung injury

## Abstract

Mechanical ventilation is usually required for saving lives in critically ill patients; however, it can cause ventilator-induced lung injury (VILI). As VEGF-secreting Ly6C^high^ monocytes are involved in VILI pathogenesis, we investigated whether cyclooxygenase-2 (COX-2) activity regulates the recruitment of VEGF-secreting Ly6C^high^ monocytes during VILI. The clinically relevant two-hit mouse model of VILI, which involves the intravenous injection of lipopolysaccharide prior to high tidal volume (HTV)-mechanical ventilation, was used in this study. To investigate the role of COX-2 in the recruitment of VEGF-secreting Ly6C^high^ monocytes during VILI, celecoxib, which is a clinical COX-2 inhibitor, was administered 1 h prior to HTV-mechanical ventilation. Pulmonary vascular permeability and leakage, inflammatory leukocyte infiltration, and lung oxygenation levels were measured to assess the severity of VILI. HTV-mechanical ventilation significantly increased the recruitment of COX-2-expressing Ly6C^high^, but not Ly6C^low^, monocytes. Celecoxib significantly diminished the recruitment of Ly6C^high^ monocytes, attenuated the levels of VEGF and total protein in bronchoalveolar lavage fluid, and restored pulmonary oxygenation during VILI. Our findings demonstrate that COX-2 activity is important in the recruitment of VEGF-secreting Ly6C^high^ monocytes, which are involved in VILI pathogenesis, and indicate that the suppression of COX-2 activity might be a useful strategy in mitigating VILI.

## 1. Introduction

Mechanical ventilation is required for patients that suffer from acute respiratory failure or acute respiratory distress syndrome (ARDS), which can be triggered by direct pulmonary insult and indirect extra-pulmonary insult [1]. However, mechanical ventilation can potentially exacerbate the condition by overly distending the regional pulmonary alveoli, particularly with large tidal volumes, which causes ventilator-induced lung injury (VILI) [2,3]. VILI may occur in healthy lungs or worsen pre-existing pulmonary disease, such as ARDS [4,5], which is characterized by up-regulating the recruitment of numerous subsets of leukocytes that overproduce inflammatory cytokines for increasing pulmonary-vasculature permeability, causing protein-rich pulmonary edema, and ultimately worsening gas exchange [6,7]. The recruitment of inflammatory leukocytes, particularly alveolar macrophages [8] and neutrophils [9,10], which migrate to the pulmonary microenvironment and release inflammatory and injurious mediators, plays a critical role in the pathogenesis of VILI [4]. Moreover, recent studies showed that various subsets of monocytes are also involved in the development of VILI [11,12,13].

The inflammatory subsets of monocytes, which are heterogeneous and pluripotent cells, consists of at least two phenotypically distinct cells that can be distinguished by Ly6C expression that serves as a specific marker for the monocytic lineage in mice [14,15,16]. Ly6C^high^CCR2^high^CX3CR1^low^ monocytes, which express high levels of Ly6C (Ly6C^high^) and C-C chemokine receptor 2 (CCR2^high^) and low levels of CX3C chemokine receptor 1 (CX3CR1^low^), are designated as the inflammatory subset, and they are derived from bone marrow macrophage-dendritic precursor cells that migrate to inflammatory sites during injury [14,15]. In contrast, the Ly6C^low^CCR2^low^CX3CR1^high^ monocytes are designated as the resident subset and they can enter tissues, irrespective of inflammation [15]. Studies have reported that the rapid recruitment of monocytes, particularly the Ly6C^high^ subset, to the lung microvasculature, contributes to pulmonary-vascular injury in a murine model of systemic endotoxemia [17,18].

Cyclooxygenases (COXs), including the constitutive COX-1, are expressed in most tissues, and the inducible COX-2 expressed in lung resident inflammatory, endothelial, and epithelial cells are responsible for catalyzing the conversion of arachidonic acid to prostanoids, which are involved in inflammation, vascular homeostasis, and vascular permeability [19]. Recent evidence has shown that COX-2 is a potent mediator of VILI progression [20,21]. The expression and activity of COX-2, but not of COX-1, are induced in lungs that were subjected to high tidal volume (HTV)-mechanical ventilation, and the expression of COX-2 is limited to alveolar monocytes and macrophages [21]. Furthermore, CAY10404, a COX-2-specific inhibitor, potently diminishes tpulmonary permeability, neutrophil infiltration, and cytokine production during VILI [21]. However, the more precise role of COX-2 expression and activity in the pulmonary monocytes, a complex and heterogenous monocytic subset, which modulate VILI, is unknown. Furthermore, studies have shown that activated Ly6C^high^ monocytes are recruited to the lung in VILI, which participate in a decrease of lung compliance [11,12]. Our recent study also demonstrates that Ly6C^high^ monocytes through enhancing vascular endothelial growth factor (VEGF) secretion for increasing pulmonary vascular permeability on causing VILI [13], indicating the critical role of the recruitment of VEGF-secreting Ly6C^high^ monocytes during VILI. Thus, we sought to investigate the functional role and therapeutic potential of COX-2 in VEGF-secreting Ly6C^high^ monocytes during VILI.

Presently, we explore the kinetic changes of COX-2-expressing Ly6C^high^ monocytes in a clinically relevant two-hit VILI mouse model. Moreover, using celecoxib, which is a COX-2-specific inhibitor, we demonstrated that COX-2 activity is important in mediating the recruitment of VEGF-secreting Ly6C^high^ monocytes that contribute to the increase of total protein in bronchoalveolar lavage fluid (BALF) during the development of VILI. Thus, the COX-2 blockade may have a therapeutic value in the treatment of VILI.

## 2. Results

### 2.1. Celecoxib, a Clinical COX-2 Inhibitor, Attenuates VILI

While a previous study has demonstrated that the expression and activity of COX-2 are critically involved in the pathogenesis of VILI [21], the detailed functional mechanism of COX-2 in Ly6C monocytes during VILI is unclear, and the therapeutic significance of celecoxib, which is a clinical COX-2 specific inhibitor, in VILI remains to be studied. To investigate the effect of the COX-2 pathway in VILI, our previously established murine model of VILI [13], which consisted of intraperitoneal administration of celecoxib (20 or 40 mg/kg) 1 h prior to HTV-mechanical ventilation, was used. In our preliminary experiments, the small-scale doses of celecoxib had been screened in mouse. We found that the doses of celecoxib that were lower than 10 mg/kg had no significant effect on VILI, and celecoxib (20 and 40 mg/kg) had better action on VILI in our system; thus, celecoxib (20 and 40 mg/kg) was applied to this study. The results (Figure 1A) showed that HTV caused significant lung injury with the infiltration of inflammatory leukocytes into the alveolus. The administration of celecoxib significantly diminished the severity of lung injury in HTV-ventilated mice (Figure 1A). Moreover, Figure 1B demonstrates that celecoxib notably reduced the infiltration of neutrophils and mononuclear cells into BALF during VILI. These results indicate the importance of COX-2 and the clinical benefit of celecoxib in treating VILI.

### 2.2. The Recruitment of COX-2-Expressing Neutrophils and Ly6C^high^, but Not Ly6C^low^, Monocytes Is Enhanced during VILI

A previous study demonstrated that HTV-mechanical ventilation induced the recruitment of COX-2-expressing mononuclear cells to the injured lung alveolus [21]. As monocytes are a subset of complex and heterogeneous cells, it is important to investigate the role of COX-2 in accurate monocytes. Accordingly, our recent study indicated that VEGF-expressing Ly6C^high^ monocytes are recruited during VILI [13]. Therefore, we aimed to determine the expression and activity of COX-2 in the recruitment of Ly6C monocytes during VILI. Thus, the time course recruitment of COX-2-expressing cells in response to the development of VILI was investigated. The staining and gating strategies of flow cytometry for identifying cell types is shown (Figure 2A,B). The low side scatter (SSC) and CD11b-positive events (P1) were gated in the analysis of monocytes with Ly6C and COX-2 expressions. COX-2-expressing Ly6C^high^ monocytes were COX-2-positive and they showed high Ly6C expression (Q2). COX-2-expressing Ly6C^low^ monocytes were COX-2-positive and revealed low Ly6C expression (Q4). A high degree of SSC and high levels of CD11b expression (P4) were used for the gating and analysis of neutrophils with Ly6C and COX-2 expressions. COX-2-expressing neutrophils were COX-2-positive and showed high Ly6C expression (Q2-1). The quantification of time course recruitment of COX-2-expressing Ly6C monocytes and neutrophils during VILI is shown (Figure 2C). The recruitment of COX-2-expressing neutrophils significantly increased by 2 h and plateaued from 4 to 6 h, but the changes in COX-2-expressing neutrophil recruitment from 2 to 4, or 6 h were not significant, indicating that COX-2-expressing neutrophil recruitment occurs early in VILI. Moreover, the recruitment of COX-2-expressing Ly6C^high^ monocytes in response to HTV-mechanical ventilation occurred later than COX-2-expressing neutrophil recruitment, increasing significantly by 4 h and peaking at 6 h. However, there was no significant change in the number of COX-2-expressing Ly6C^low^ monocytes during these time points. These data revealed the critical involvement of COX-2-expressing Ly6C^high^ monocytes in the progression of VILI (Figure 2). Furthermore, when compared to our previous publication [19], it is worth noting that the tendency of the time course recruitment of COX-2-expressing Ly6C^high^ monocytes is similar to the VEGF-expressing Ly6C^high^ monocytes during VILI development on VEGF production and pulmonary-vasculature leakage, suggesting that COX-2 activity might associate with VEGF secretion in the Ly6C^high^ monocytes for causing VILI.

### 2.3. Celecoxib Aignificantly Mitigates the Recruitment of Ly6C^high^, but Not Ly6C^low^, Monocytes in VILI

Although COX-2 inhibition potently restrains VILI [21], the functional effect of a COX-2 signaling blockade on the recruitment of Ly6C^high^ monocytes in VILI is unknown. Figure 3A shows the flow cytometric technique for quantifying the recruitment of specific Ly6C monocytes in response to COX-2 inhibition. Low SSC and CD11b-positive events (P1) were gated to analyze Ly6C and F4/80 expression. Ly6C^high^ monocytes were F4/80-positive and they showed high Ly6C expression (P2). Ly6C^low^ monocytes were F4/80-positive and showed low Ly6C expression (P3). Figure 3B shows the quantification of Ly6C^high^ and Ly6C^low^ monocytes. The number of Ly6C^high^, but not Ly6C^low^, monocytes was higher in the HTV-injured lung than in the lungs of control, non-ventilated mice. Celecoxib had no significant effect on the recruitment of Ly6C^low^ monocytes (Figure 3B). However, celecoxib treatment diminished the recruitment of Ly6C^high^ monocytes in a dose-dependent manner (Figure 3B), indicating the importance of COX-2 in mediating the recruitment of Ly6C^high^ monocytes during VILI.

### 2.4. Celecoxib Reduces Alveolar Protein Outflow and VEGF Secretion into Bronchoalveolar Lavage Fluid (BALF) and Improves Pulmonary Oxygenation in VILI

The total quantity of proteins and VEGF in BALF are an indicator of pulmonary-vasculature permeability [13,22]. Assaying total proteins and VEGF secretion on reflecting pulmonary-vasculature leakage during VILI evaluated the therapeutic effect of celecoxib. Figure 4A,B show that VEGF and total protein levels in BALF were significantly increased with HTV-mechanical ventilation. Celecoxib treatment meaningfully blunted the increase of total protein outflow and VEGF secretion in BALF after VILI in a dose-dependent manner (Figure 4A,B). Moreover, an analysis of cardiac blood samples to investigate the status of pulmonary oxygenation showed that PaO_2_ (Table 1) and the PaO_2_/FIO_2_ ratio (Figure 4C) were low in HTV-mechanical ventilation, indicating acute lung injury and hypoxemia. However, the level of pulmonary oxygenation was preserved by celecoxib treatment after HTV-mechanical ventilation. No significant differences in pH, PaCO_2_, or HCO_3_ were found among the four experimental groups (Table 1), indicating that there was no severe metabolic acidosis during the experiments. This data reveals that celecoxib exerts a therapeutic benefit in VILI by preventing pulmonary-vasculature damage and preserving pulmonary oxygenation.

### 2.5. VEGF Secretion in Ly6C^high^ Monocytes Is COX-2-Dependent

The recruitment of VEGF-secreting Ly6C^high^ monocytes plays an important role in the pathogenesis of VILI [13]. However, it is unclear whether COX-2 activity directly regulates VEGF secretion in Ly6C^high^ monocytes on VILI. The Ly6C^high^ monocytes were sorted from HTV-injured lungs for testing the involvement of COX-2 on VEGF secretion ex vivo. Figure 5 shows that VEGF production by sorted Ly6C^high^ monocytes was significantly increased 6 h after culture (Figure 5). However, this increment of VEGF secretion by the sorted Ly6C^high^ monocytes was considerably suppressed in the presence of celecoxib (10 µM). This result denotes the importance of COX-2 and VEGF functional axis in Ly6C^high^ monocytes during the development of VILI, and it provides the clinical insights of celecoxib on targeting COX-2-expressing and VEGF-secreting Ly6C^high^ monocytes for treating VILI in patients.

## 3. Discussion

Mechanical ventilation is used as supportive therapy for patients with acute respiratory failure or ARDS. However, due to the morphological variation of patients’ lungs, even when protective ventilation is used, and the uneven overdistention of regional alveoli, which leads to the exacerbation of VILI [23], suggests that more detailed mechanistic and pharmacologic investigations are required in identifying the therapeutic window for improving the clinical outcomes in ventilated patients. By using our previous established VILI murine model [13], here we demonstrate that COX-2 activity is critically important in the recruitment of functional, VEGF-secreting Ly6C^high^, but not Ly6C^low^, monocytes; this contributes to the pathogenesis of VILI via enhancing pulmonary-capillary permeability and -vasculature leakage in reducing pulmonary oxygenation. Celecoxib, which is a clinically used COX-2 inhibitor that affects functional COX-2 and VEGF expressions in Ly6C^high^ monocytes exhibits therapeutic value in treating and/or preventing VILI in patients.

Neutrophils and monocytes are the important cell types in inflammation and VILI [9,11]. In our VILI model, the COX-2-expressing neutrophils were significantly recruited into HTV-injured lungs after 2 h and recruitment had plateaued by 4 to 6 h. Moreover, COX-2-expressing Ly6C^high^ monocytes were not significantly elevated after 2 h, increased by 4 h, and then significantly increased at 6 h At 0 and 2 h, there were fewer COX-2-expressing Ly6C^high^ monocytes than neutrophils; at 4 h, there were equivalent numbers of COX-2-expressing monocytes and neutrophils; and at 6 h, there were more COX-2-expressing monocytes than neutrophils. This result indicated that the COX-2-expressing neutrophils are the first cells to arrive in the ventilator-injured lung and they may employ paracrine signaling to recruit the wave of COX-2-expressing Ly6C^high^ monocytes that further amplify the acute phase of lung-injured cascade. Since the role of neutrophils in the process of lung injury has been well-investigated [9], the importance of COX-2 expression in mononuclear cells is established [21]. As the COX-2-specific inhibitor CAY10404 and parecoxib can potently diminish VILI [21,24], the precise role of COX-2 activity and expression in Ly6C^high^ monocytes was therefore focused on the pathogenesis of VILI in this study.

Ly6C^high^CCR2^high^CX3CR1^low^ monocytes have been demonstrated to be recruited into inflamed tissues through the activation of CCR2, which is a receptor of chemokine C-C-chemokine ligand 2 (CCL2) [25]. However, the underlying mechanism of the recruitment and activity of Ly6C^high^ monocytes in VILI is unknown. Moreover, how COX-2-activated prostaglandins or other mediators, including cytokines, chemokines, and growth factors, mobilize and activate Ly6C^high^ monocytes in the pathogenesis of VILI remains unclear. CCL2 is required in guiding the migration of CCR2-expressing monocytes into inflamed tissues [26], and CCL2 has been shown to accumulate in BALF in a model of LPS-induced VILI [27]. Further, the induction of COX-2 is required for CCL2 expression by monocytes during the progression of atherosclerosis [2]. In CCR2 knockout mice, the accumulation of Ly6C^high^ monocytes in injured alveoli is suppressed, protecting the mouse from fibrosis [28]. Moreover, the plasma and BALF levels of interleukin-1β (IL-1β), a specific inducer of COX-2 expression [29], are associated with clinical VILI, and the blockade of IL-1β ameliorates VILI [30]. Here, we show that COX-2 inhibition inhibited the recruitment of Ly6C^high^ monocytes that have been reported to express CCR2 [31], resulting in the mitigation of VILI. These findings indicate the activation of COX-2 from a wide variety of HTV-activated resident cells in the lung for regulating the recruitment and activation of Ly6C^high^ monocytes. During VILI, COX-2 activation by proinflammatory mediators may stimulate the secretion of CCL2 to govern the lung margination of inflammatory cells, such as Ly6C^high^ monocytes. Moreover, these findings contribute therapeutic insights into the inhibitory effect of celecoxib on the IL-1β-COX-2-CCL2 signaling axis, which suppresses the mobilization of Ly6C^high^ monocytes, thereby attenuating VILI.

Further, we show that the inhibition of COX-2 activity by celecoxib can significantly diminish the VEGF and BALF total protein levels. It is known that there is a high correlation between the expression of COX-2 and VEGF, and that NS-398, a COX-2 inhibitor, inhibits VEGF expression in cancer cells [32]. Additionally, celecoxib reduces the transcriptional activity of Sp1, which down-regulates VEGF expression by cancer cells [33]. Our recent study also determined that the Ly6C^high^ monocytes enhance VEGF expression for increasing pulmonary vascular permeability, which leads to VILI [13]. Presently, our data further showed that Ly6C^high^ monocytes time-dependently secreted VEGF, and this VEGF secretion was inhibited by celecoxib, indicating that VEGF secretion by Ly6C^high^ monocytes during VILI is COX-2-dependent. Furthermore, these results reveal the importance of COX-2 inhibition by clinical COX-2 inhibitors that are used for treating and/or preventing VILI, via the interruption of the COX-2-VEGF functional axis in Ly6C^high^ monocytes.

Here, we focused on the COX-2-expressing Ly6C^high^ monocytes, but the role of COX-2-expressing neutrophils in VILI cannot be ignored, because COX-2-expressing neutrophils and Ly6C^high^ monocytes may sequentially interplay with each other during VILI. A previous study showed that a COX-2 inhibitor potently inhibits the recruitment of neutrophils during VILI [21]. In our study, mononuclear cells and neutrophils were accumulated in HTV-injured lungs, and this effect was also significantly blocked by celecoxib treatment. Therefore, the inhibition of COX-2 activity may provide a wide range of therapeutic effects, such as the inhibition of early recruitment of COX-2-expressing neutrophils, thereby reducing the recruitment of VEGF-secreting Ly6C^high^ monocytes in late to prevent the VEGF-stimulated pulmonary permeability increment. A previous study showed that the expression of induced form of COX-2 is relatively low in mouse lung tissue, but it is strongly upregulated after ventilation, especially HTV for 4 h, indicating an acute phase injury of ventilation on upregulating COX-2 expression. Furthermore, the COX-2 expressing cells in ventilation-injured lung were verified using immunohistochemically staining, which show a significant positivity staining of pulmonary alveolar and interstitial mononuclear cells (CD45 and CD68 positive monocyte/macrophage lineage), rather than bronchiolar epithelium on COX-2 expression [21]. Accordingly, our result was consistent with this study and it provided more detailed information that the subset of mononuclear cells, Ly6C^high^-COX-2-expressing monocytes, are critically involved in VEGF secretion during VILI. Due to pulmonary epithelial and endothelial cells being evidenced to express COX-2 after various stimulations on resulting lung injury [19,34,35,36], therefore, we cannot exclude the involvement of COX-2 on pulmonary epithelial and endothelial cells for contributing to exacerbate VILI in the late phase other than the early acute phase. However, the temporal, reciprocal, and sequential expression of COX-2 between inflammatory leukocyte subsets, alveolar epithelium, and pulmonary endothelium during VILI development are unknown and are worthy of further investigation.

COX-2 acts as a key role in arachidonic acid metabolism in both physiological and pathological conditions [37]. It is constitutively expressed in human kidney and brain tissues and is inducible in various cells, including monocytes/macrophages, endothelial cells, epithelial cells, and cancer cells during stimulations of inflammatory cytokines, laminar shear stress, growth factors, and VILI [21,38]. Furthermore, the activity of COX-2 on generating prostaglandin E2 (PGE2) and prostacyclin are involved in various biological processes, including renal hemodynamics, the control of blood pressure, endothelial thromboresistance, pain and inflammation, and VILI [21,38]. Hence, it is predictable that COX-2 inhibition by COX-2 inhibitors, such as celecoxib, may exhibit multifaceted clinical outcomes other than diminishing VILI, ranging from increasing cardiovascular hazard [39], blood pressure [40], and atherothrombotic risk [41]. However, due to patients’ variations and disease status, the depiction of the practical consequences of COX-2 inhibition in humans has been problematic and has generated many controversial findings [42]. Besides, stroke volume and cardiac output reductions may alter the ventilation/perfusion ratio in the lungs and compromise the gas exchange [43]. In our experimental system, the cardiovascular hazards, such as hypoxemia and metabolic acidosis, were not found in different doses of celecoxib (20 and 40 mg/kg) administration during a short period. Even though, it is unclear whether a long period or higher doses of celecoxib administration will affect cardiovascular-related functions. Therefore, the long-term application of the COX-2 inhibitor should be very carefully evaluated, especially on the cardiovascular hazard issues.

## 4. Materials and Methods

### 4.1. Animals

Six to eight-week-old, C57BL/6 male mice (20–25 g) were purchased from the National Laboratory Animal Center (Taipei, Taiwan). All of the animal experiments were conducted according to the National Institutes of Health guidelines (Guides for the Care and Use of Experimental Animals) that Institutional Animal Care and Use Committee of Chang Gung Memorial Hospital, Chiayi, Taiwan approved (Approval number: 2013070301 (23 October 2013) and 2017030202 (15 May 2017)).

### 4.2. Experimental Model of Ventilation-Induced Lung Injury

We used our previously established murine model of VILI [13] for studying the molecular mechanism of COX-2 on Ly6C^high^ monocytes in VILI. Celecoxib (20 or 40 mg/kg; Sigma-Aldrich, St. Louis, MO, USA) was intraperitoneally administered 1 h prior to mechanical ventilation. Following this, lipopolysaccharide (LPS; O111B4; Sigma-Aldrich) (20 ng per mouse) was intravenously administered, immediately prior to mechanical ventilation. After tracheostomy, a 20-gauge angiocatheter was introduced into the tracheostomy orifice under general anesthesia by intraperitoneal injection with Zoletil 50 (80 mg/kg; Tiletamine-Zolazepam, Virbac, Carros, France). Anesthesia was maintained with an intraperitoneal injection of Zoletil 50 (10 mg/kg/h) during mechanical ventilation. The mice were placed in a supine position on a heated blanket and then attached to a specialized rodent ventilator (SAR-830/AP; CWE Inc., Ardmore, PA, USA) that was ventilated with HTV (VT 20 mL/kg, 60 breaths/min) for 6 h while the mice were breathing room air with zero end-expiratory pressure. At the end of experiment, the mice were sacrificed via an overdose of anesthetic. The non-ventilated mice served as control. Terminal blood samples of mice were obtained by cardiac puncture and were used for blood gas analysis to assess pulmonary oxygenation and the severity of acute lung injury [44].

### 4.3. Analysis of BALF

At the end of the experiment, the catheter that had been inserted into the trachea was lavaged three times with 1 mL of phosphate buffered saline to collect BALF. BALF was centrifuged at 300 × *g* for 10 min and the supernatant was collected for a measurement of total protein and VEGF levels. Furthermore, the total number of leukocytes in the sediment was calculated while using a hemocytometer (Paul Marienfeld GmbH, Lauda-Koenigshofen, Germany). A 100 μL aliquot of BALF sediment was further smeared on a glass slide for staining with Liu’s stain, which can differentiate and quantify the leukocyte subsets [13]. The VEGF level in BALF was measured by enzyme-linked immunosorbent assays (R&D Systems, Minneapolis, MN, USA). BALF total protein was determined while using a Pierce protein assay kit (Pierce, Rockford, IL, USA). Both of the analyses were conducted according to the manufacturer’s instructions.

### 4.4. Classification Strategy of Flow Cytometry

As described previously [13], the mouse lung homogenate was processed to produce a single cell suspension for flow cytometry. Briefly, lungs that were excised from each group of mice were mechanically disrupted and prepared as single cell suspensions by passing through a 40-μm strainer (BD Biosciences, San Jose, CA, USA). The pulmonary single cell suspension was stained using fluorophore-conjugated anti-mouse antibodies (eBioscence, San Diego, CA, USA) against CD11b (Clone M1/70, anti-mouse CD11b FITC; 1:200), Ly6C (Clone HK1.4, anti-mouse Ly6C APC; 1:200), F4/80 (Clone BM8, anti-mouse F4/80 PE; 1:300), or an appropriate isotype-matched control. To analyze the intracellular proteins, the pulmonary single cell preparation was permeabilized while using Cytofix/Cytoperm solution and Perm/Wash Buffer (BD Biosciences). The pulmonary single cell preparation was further stained with fluorophore-conjugated anti-mouse antibody (Santa Cruz Biotechnology, Santa Cruz, CA, USA) against COX-2. Fluorescence was determined using FACSCanto II and analyzed with FACSDiva software version 7.0 (BD Biosciences). The viable Ly6C^high^ monocytes were sorted using the FACSAria Fusion cell sorter and analyzed with FACSDiva software version 8.0 (BD Biosciences).

### 4.5. Ex vivo VEGF Secretion Assay

Ly6C^high^ monocytes that were sorted from the HTV-injured lungs were seeded (1 × 10^5^ cells per well) in the wells of a 96-well plate and cultured in an RPMI medium containing 10% fetal bovine serum in the presence or absence of celecoxib (10 μM). After 1 and 6 h of culture, Ly6C^high^ monocyte-conditioned medium was separately collected for measuring the secretion of VEGF by Ly6C^high^ monocytes while using an enzyme-linked immunosorbent assay (R&D Systems).

### 4.6. Statistical Analysis

All the data are expressed as the mean plus or minus the standard deviation (SD). Statistical significance was examined using unpaired Student’s *t*-test. Comparisons among multiple groups were made by one-way analysis of variance with Bonferroni-corrected pairwise post hoc comparisons. All of the statistical analyses were performed with GraphPad Prism software version 5.0 (GraphPad Software, La Jolla, CA, USA). A *p*-value of <0.05 was considered to be statistically significant.

## 5. Conclusions

The present study defines the time course and mechanistic significance of a COX-2-dependent mechanism for the recruitment of activated, VEGF-secreting Ly6C^high^ monocytes, which increase the pulmonary-vasculature permeability and contribute to the pathogenesis of a clinically relevant murine model of VILI with HTV-mechanical ventilation. Therefore, the adjuvant pharmacologic application of COX-2 inhibitors to target the recruitment of COX-2- and VEGF-expressing Ly6C^high^ monocytes might provide novel therapeutic insights that will facilitate the treatment and/or prevention of clinical VILI.

## Figures and Tables

**Figure 1 ijms-20-01771-f001:**
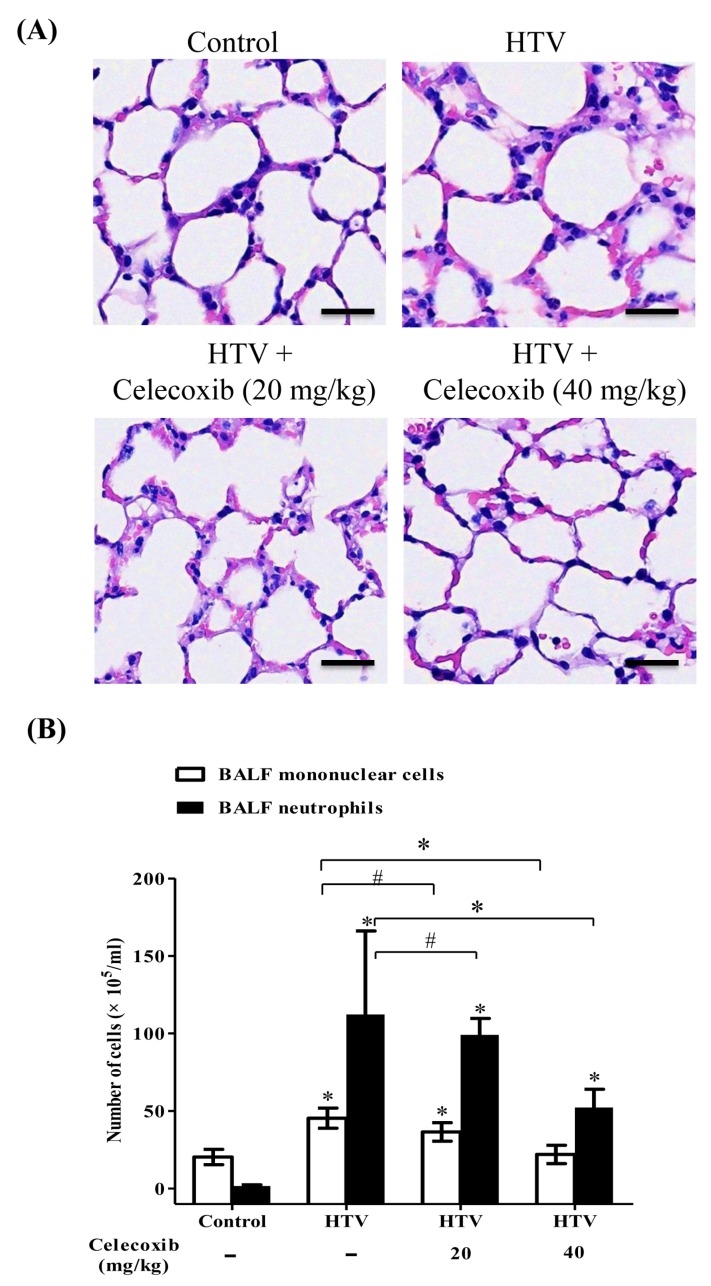
Celecoxib attenuated ventilator-induced lung injury (VILI) and decreased the infiltration of leukocytes in BALF during VILI. (**A**) Representative micrographs of hematoxylin-and-eosin-stained sections of the lung. Scale bar = 100 μm. (**B**) Cytological analyses of mononuclear cells and polymorphonuclear neutrophils (PMN) in bronchoalveolar lavage fluid (BALF). Values represent the mean ± SD (*n* = 6). ^#^
*p* < 0.05 and * *p* < 0.01, when compared with the control or between groups.

**Figure 2 ijms-20-01771-f002:**
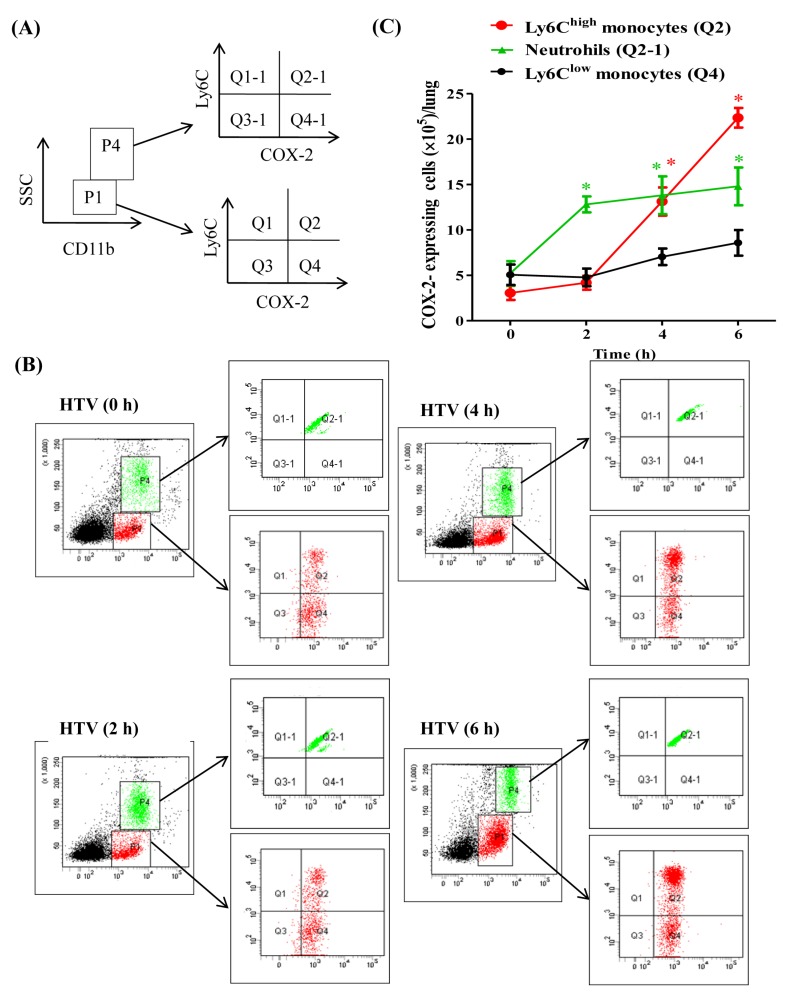
Cyclooxygenase-2 (COX-2)-expressing Ly6C^high^ monocytes recruited into the lung during VILI. (**A**,**B**) Gating strategy of flow cytometry analysis. (**C**) Time course recruitment of COX-2-expressing Ly6C^high^ monocytes, Ly6C^low^ monocytes, and neutrophils during VILI. Values represent the mean ± SD (*n* = 6). * *p* < 0.01 as compared with the control at that time point.

**Figure 3 ijms-20-01771-f003:**
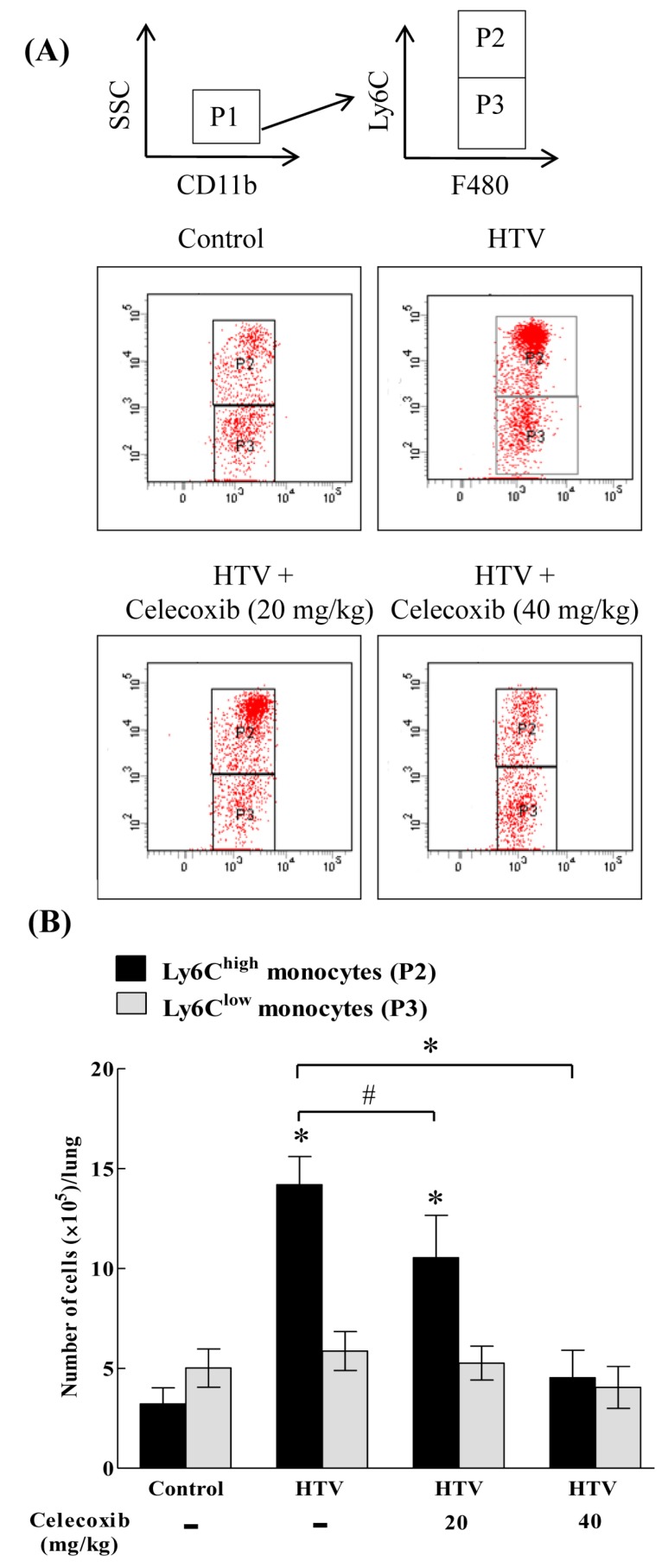
Celecoxib significantly diminished the recruitment of Ly6C^high^, but not Ly6C^low^, monocytes in VILI. (**A**) Gating strategy of flow cytometry analysis. (**B**) The numbers of Ly6C monocytes in lung homogenates. Values represent the mean ± SD (*n* = 6). ^#^
*p* < 0.05 and * *p* < 0.01, when compared with the control or between groups.

**Figure 4 ijms-20-01771-f004:**
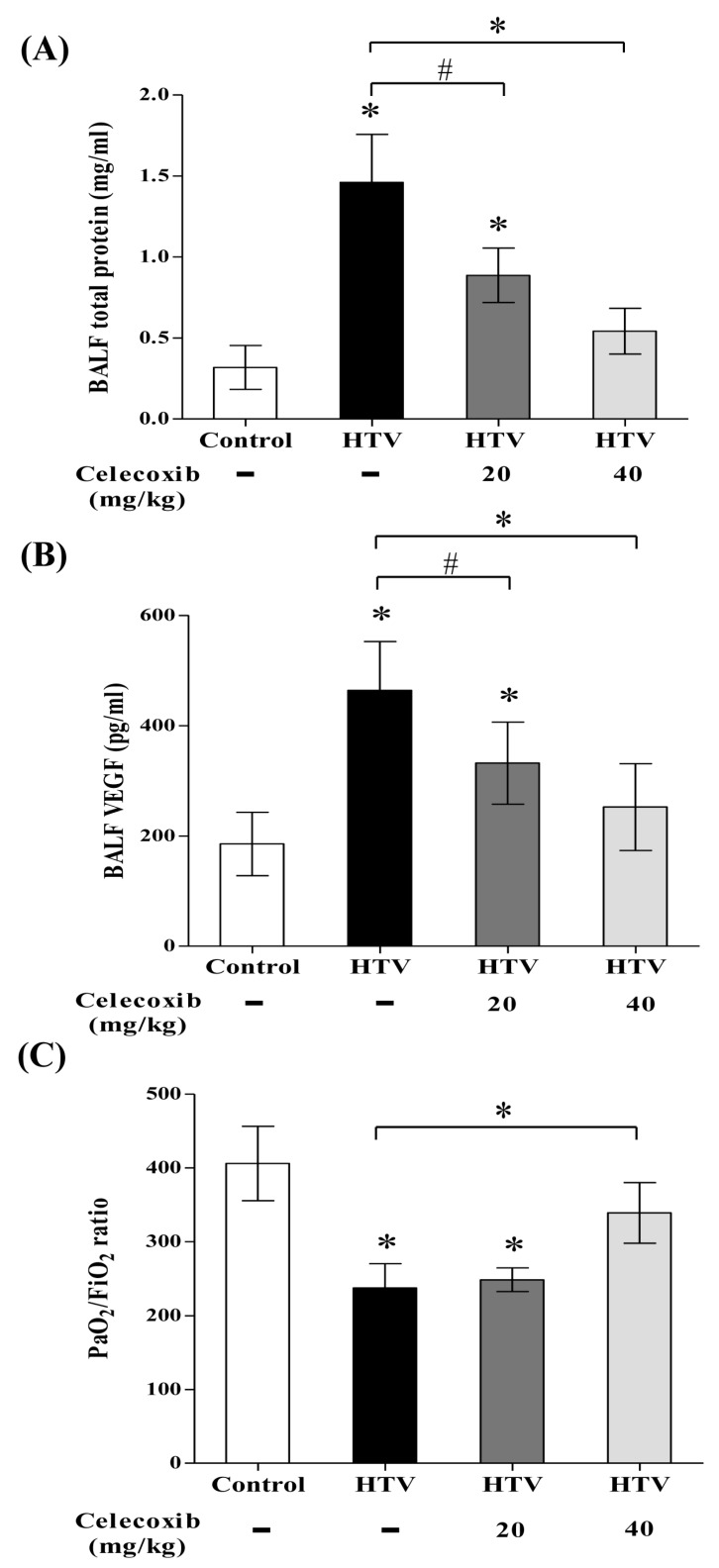
Celecoxib significantly reduced pulmonary-vasculature permeability and improved pulmonary oxygenation in VILI. (**A**) BALF total protein. (**B**) BALF vascular endothelial growth factor (VEGF). (**C**) PaO_2_/FiO_2_. Values represent the mean ± SD (*n* = 6). ^#^
*p* < 0.05 and * *p* < 0.01, as compared with the control or between groups.

**Figure 5 ijms-20-01771-f005:**
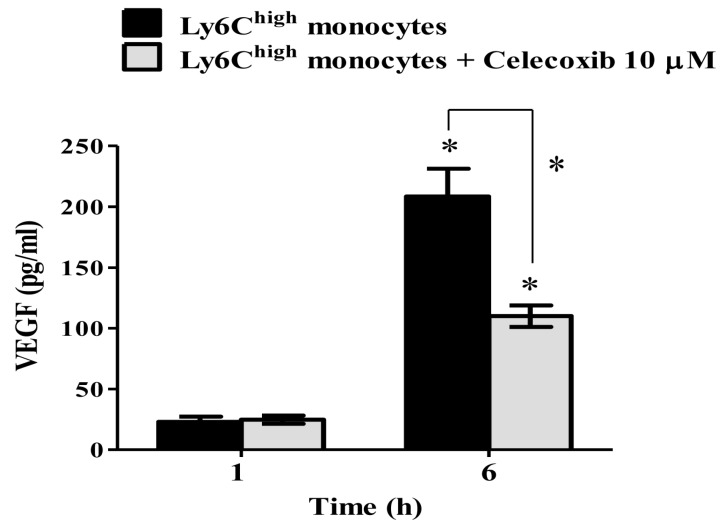
The secretion of VEGF by sorted Ly6C^high^ monocytes was COX-2-dependent. Values represent the mean ± SD (*n* = 4). * *p* < 0.01 compared to individual group of 1 h or compared with group untreated with celecoxib.

**Table 1 ijms-20-01771-t001:** Analysis of cardiac blood gas.

	Control	HTV	HTV	HTV
Celecoxib (mg/kg)	—	—	+20	+40
pH	7.371 ± 0.031	7.322 ± 0.026	7.307 ± 0.02	7.37 ± 0.025
PaCO_2_ (mm Hg)	35.83 ± 5.56	30.5 ± 3.72	34.9 ± 4.23	36.23 ± 2.15
HCO_3_ (mmol/l)	21.48 ± 1.75	17.51 ± 1.30	18.8 ± 0.89	21.3 ± 1.45
PaO_2_ (mm Hg)	82.83 ± 11.28	51.0 ± 6.89 *	52.3 ± 3.31	71.21 ± 8.6 ^#^

Values represent the mean ± SD (*n* = 6). * *p* < 0.05 compared with control group. ^#^
*p* < 0.05 compared with HTV group.

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
