# Peer review of "Cyclooxygenase-2 Activity Regulates Recruitment of VEGF-Secreting Ly6Chigh Monocytes in Ventilator-Induced Lung Injury"

_ijms, 2019, doi:10.3390/ijms20071771_

Reviewer 1 Report

The manuscript by Huang et al., reported that COX-2 activity is important for the recruitment of VEGF-secreting LY6C monocytes which are involved in VILI pathogenesis.

Despite the work is well conducted and the results are quite interesting, there some concerns to be addressed:

i) What is the cellular source of COX-2 activity in VILI pathogenesis? Are inflammatory cells and/or tissue cells? Please explain.

ii) The dose of celecoxib administered (20 and 40 mg/kg) was appropriate to inhibit COX-2 activity in vivo? How this dose was chosen?

iii) The authors suggested COX-2 inhibition as a possible therapeutic approach; please discuss the fact the coxib administration is associated with the cardiovascular hazard.

iv) To determine whether celecoxib treatment in vitro affected COX-2 activity, I would suggest measuring prostaglandins levels (such as PGE2) in the BALF supernatant. 

Author Response

Dear Reviewer,

Thank you for the comments

We have added aforesaid explanation in author-coverletter-4060664.

Best regards

Shi, Chung-Sheng 
Ph.D. Assistant professor 
Graduate Institute of Clinical Medical Sciences
College of medicine
Chang Gung University 
Tel: +886-5-3621000 ext.2100 (405-2100)

Reviewer 2 Report

The manuscript ijms-474263 ("Cyclooxygenase-2 activity regulates recruitment of VEGF-secreting Ly6C high monocytes in ventilator-induced lung injury") for IJMS is very interesting. In my opinion, the study is of scientific relevance, gives new information and should be published in IJMS. Moreover this manuscript has high impact, a significant addition to the literature of the field, provide novel therapeutic insights. Comments on the paper: This paper is very well written and clear.  Manuscript is properly organized. Title and highlights reflects well the content of manuscript. The abstract reflects the content of manuscript. Introduction is well written and the goal is clearly formulated.  Results, discussion and the conclusions are Well written. The paper is clear and succinct. References do not meet the editorial requirements of IJMS (references should be checked, authors used sometimes full names of articles or only abbreviations= should be abbreviations. Table is clear and easy to follow, but should have a higher sharpness. Figures and descriptions of legend are clear and understandable.

Author Response

Dear Reviewer,

Thank you for the comment and suggestion. We have revised the format accordingly.

Best regards

Shi, Chung-Sheng 
Ph.D. Assistant professor 
Graduate Institute of Clinical Medical Sciences
College of medicine
Chang Gung University 
Tel: +886-5-3621000 ext.2100 (405-2100) 
